# Novel Pathogenic Variants in Hereditary Cancer Syndromes in a Highly Heterogeneous Cohort of Patients: Insights from Multigene Analysis

**DOI:** 10.3390/cancers16010085

**Published:** 2023-12-23

**Authors:** Airat Bilyalov, Anastasiia Danishevich, Sergey Nikolaev, Nikita Vorobyov, Ivan Abramov, Ekaterina Pismennaya, Svetlana Terehova, Yuliya Kosilova, Anastasiia Primak, Uglesha Stanoevich, Tatyana Lisica, German Shipulin, Sergey Gamayunov, Elena Kolesnikova, Igor Khatkov, Oleg Gusev, Natalia Bodunova

**Affiliations:** 1Institute of Fundamental Medicine and Biology, Kazan Federal University, 420008 Kazan, Russia; 2SBHI Moscow Clinical Scientific Center Named after Loginov MHD, 111123 Moscow, Russiai.hatkov@mknc.ru (I.K.);; 3The Federal State Budgetary Scientific Institution “Izmerov Research Institute of Occupational Health”, 105275 Moscow, Russia; 4Ministry of Health Kursk Region, 305000 Kursk, Russia; 5Kursk Regional Scientific and Clinical Center Named after G. Y. Ostroverkhov, 305524 Kursk, Russia; s.terechova1972@mail.ru (S.T.); kosilova.92@inbox.ru (Y.K.); a.primak94@yandex.ru (A.P.);; 6Centre for Strategic Planning and Management of Biomedical Health Risks, Federal Medical and Biological Agency, 119435 Moscow, Russia; 7Nizhny Novgorod Regional Oncologic Hospital, 603163 Nizhny Novgorod, Russia; 8Life Improvement by Future Technologies (LIFT) Center, 121205 Moscow, Russia

**Keywords:** NGS, cancer, multigene panel

## Abstract

**Simple Summary:**

This study addresses the global healthcare challenge of cancer by investigating hereditary cancer syndromes (HCS) and their genetic underpinnings. Using a multigene hereditary cancer panel, we examined Russian patients with suspected HCS, revealing that 21.6% had pathogenic or likely pathogenic genetic variants. Predominant mutations were found in *BRCA1/BRCA2*, *CHEK2*, and *ATM* genes, and we identified 16 previously undescribed variants in *MUTYH*, *GALNT12*, *MSH2*, *MLH1*, *MLH3*, *EPCAM*, and *POLE* genes. Our findings underscore the importance of comprehensive genetic testing for personalized cancer prevention and treatment. This research contributes essential genetic insights, particularly in regions like Russia where epidemiological data are limited, establishing the way for improved understanding and management of hereditary cancer syndromes.

**Abstract:**

Cancer is a major global public health challenge, affecting both quality of life and mortality. Recent advances in genetic research have uncovered hereditary cancer syndromes (HCS) that predispose individuals to malignant neoplasms. While traditional single-gene testing has focused on high-penetrance genes, the past decade has seen a shift toward multigene panels, which facilitate the analysis of multiple genes associated with specific HCS. This approach reveals variants in less-studied gene regions and improves our understanding of cancer predisposition. In a study composed of Russian patients with clinical signs of HCS, we used a multigene hereditary cancer panel and revealed 21.6% individuals with pathogenic or likely pathogenic genetic variants. *BRCA1/BRCA2* mutations predominated, followed by the *CHEK2* and *ATM* variants. Of note, 16 previously undescribed variants were identified in the *MUTYH*, *GALNT12*, *MSH2*, *MLH1*, *MLH3*, *EPCAM*, and *POLE* genes. The implications of the study extend to personalized cancer prevention and treatment strategies, especially in populations lacking extensive epidemiological data, such as Russia. Overall, our research provides valuable genetic insights that give the way for further investigation and advances in the understanding and management of hereditary cancer syndromes.

## 1. Introduction

Cancer is a global unresolved healthcare challenge that is currently responsible for a significant decreased quality of life and mortality worldwide [1]. In recent years, significant progress has been made in understanding the genetic reasons for cancer development. One of the most important discoveries is the identification of hereditary cancer syndromes (HCS) characterized by an increased predisposition to the development of malignant neoplasms [2].

Traditionally, the identification of HCS relied on targeted single-gene testing, an approach primarily focused on well-established high-penetrance genes associated with specific cancer types. For example, when hereditary breast and ovarian cancer is suspected, testing for the *BRCA1/BRCA2* genes is used effectively in clinical practice [3]. However, many hereditary cancer syndromes have similar clinical symptoms, which create difficulties in their differential diagnosis.

In the last decade, multigene panels have begun to be actively used in medical genetics for diagnosing diseases. This approach allows the simultaneous analysis of multiple genes whose pathogenic variants lead to the development of a specific hereditary cancer syndrome [4,5,6]. These panels can also reveal genetic variants in less-explored regions of genes, including noncoding intronic regions, potentially elucidating previously unrecognized contributors to cancer predisposition [7,8,9]. 

Furthermore, the application of hybrid targeted gene panels enables the simultaneous assessment of various characteristics, including the identification of germline variants. These panels provide the capability to determine the methylation status of oncogenes, detect gene fusions, assess tumor mutational burden (TMB), and evaluate microsatellite instability (MSI). The research focused on multigene panels is gaining traction and actively being implemented within the medical systems of several countries. This implementation allows for a comprehensive analysis of multiple genetic features in a more efficient and integrated manner [10,11,12,13]. Brian H. Shirts et al. and Holly LaDuca et al. have previously investigated the outcomes of employing multigene panels in cancer studies, establishing a foundation for exploring the genetic landscape of cancer via comprehensive gene panels [14,15]. In this study, we build upon their approach by extending our investigation to the relatively unexplored Russian population, with the goal of providing valuable insights to the field.

Understanding the genetic basis of HCS should inform the choice of strategies for clinical observation in the long-term. This has the potential to improve the prevention of cancer recurrence and secondary tumor development. Identification of genetic variants associated with HCS is particularly important in countries like Russia where there is a lack of population-based epidemiologic studies. Such studies not only expand our knowledge of the genetic epidemiology of HCS, but also provide a better genotype–phenotype correlation understanding of a certain hereditary cancer. 

In our research, we performed genetic testing on Russian individuals with clinical evidence of HCS and/or a family history of cancer. The primary objective was to gain insights into the prevalence of various clinically significant genetic variants among cancer patients.

## 2. Materials and Methods

The study cohort included 657 patients from Russia divided into two groups: 632 (96.2%) cancer patients with clinical signs of cancer and 25 (3.8%) patients with benign tumors. These individuals were selected based on criteria established in a prior study and underwent consultations with geneticists [16]. Participation in the study involved molecular genetic testing, for which all participants provided detailed information regarding their personal and familial cancer histories. Additionally, they consented to the use of their anonymized data for the research and academic purposes. For the testing process, each participant contributed two blood samples, with each sample collected in an EDTA tube of 5 mL capacity.

To design the panel, we selected a total of 44 genes known to be involved in hereditary tumor syndromes. The full panel is presented in Appendix A. The panel was designed using the HyperDesign online service provided by Roche (Roche, Basel, Switzerland)), which incorporates the coding regions, splicing sites, and 5’-UTR regions of the selected genes into the probe design.

For sample preparation, DNA was isolated from lymphocytes using the QIAamp DNA Blood Mini Kit from Qiagen (Hilden, Germany). Library preparation was performed using the KAPA HyperPrep Kit from Roche, following the standard protocol. The prepared libraries were then subjected to hybridization with the custom panel according to the Hyper protocol from Roche.

Sequencing was performed on the MiSeq platform from Illumina (San Diego, CA, USA) using the MiSeq Reagent Kit v2 with 500 cycles, achieving coverage of up to 1000×. This allowed for the simultaneous analysis of up to 96 libraries in a single run, ensuring efficient and high-throughput sequencing of the captured coding sequences.

Paired-end reads were aligned against the reference genome (hg38) using the BWA-MEM2 algorithm [17]. Following this, duplicate sequences were identified and removed via Picard MarkDuplicates [18]. Subsequently, recalibration of base quality scores and the identification of genetic variants were performed using Genome Analysis Toolkit (GATK) tools: BQSR for score recalibration and HaplotypeCaller for variant calling [19]. The uniformity of base coverage exceeded 98% for all samples All the samples with mean coverage ≤70× were excluded from the study. Germline variants were reported if they passed all the HaplotypeCaller filters and the total number of reads covering it was ≥40. Annotation and interpretation of all identified variants were carried out using proprietary software, which utilizes interpretation standards and guidelines of the American College of Medical Genetics and Genomics and the Association of Molecular Pathology [20]. In this study, we primarily focused on pathogenic and likely pathogenic genetic variants. 

Overall, this targeted sequencing approach using a custom panel of probes and the MiSeq Illumina platform provided comprehensive coverage of the coding regions in the selected genes, enabling accurate and efficient analysis of potential genetic variants associated with tumor development [21].

## 3. Results

In our comprehensive analysis, we observed that 21.6% (142 out of 657) of the total participants had pathogenic (P) or likely pathogenic (LP) genetic variants, as outlined in Table 1. The mean age of manifestation in our study was 44.5 ± 11 years. Additionally, we observed that among the individuals with pathogenic or likely pathogenic genetic variants, there were 26 males and 116 females, providing valuable insights into the gender distribution of these variants within our study cohort. 

Full information is presented in Appendix A: Frequency of pathogenic and likely pathogenic variants.

Notably, the majority of these mutations (56, representing 39.4%) was identified in the *BRCA1*/*BRCA2* genes, primarily associated with breast (26.7%) and ovarian cancer (8.4%) syndromes. In second place were variants of *CHEK2* (14, 9.8%), which associated with breast cancer (7.7%). *ATM* (9, 6.3%), the third most common variant, was found in pancreatic (2.1%) and breast cancer (3.5%). All the variants are presented in Table 1 and Figure 1. We identified (16, 11.2%) variants that have not been found in any published studies according to the genomic databases, and are also absent from the gnomAD genomes (Table 2).

Most identified variants in our study were frameshift variants with 63 (44.4%) cases, followed by nonsense and missense variants, comprising 23.9% and 19.7%, respectively. Splicing genetic variants were found in 13 cases and accounted for 9.2% of all identified genetic alterations.

The median age for cancer diagnosis across the *BRCA1/2* mutation carriers was 46 years, in the *CHEK2* group it was 42.5 years, 44.8 years for ATM, 48.6 years for *PALB2*, 44 years for *MUTYH*, and 45.6 years for *BLM.*

Single mutations were identified in *MEN1* for gastric cancer, *MSH6* for colorectal cancer, *CDKN2A* for pancreatic cancer, and *EPCAM/TSC2/GALNT12* for breast cancer.

## 4. Discussion

The use of multigene panels in the genetic analysis of hereditary cancer syndromes offers significant advantages, especially in diverse populations such as those in Russia. A major advantage is the comprehensive inclusion of the diverse genetic landscape inherent in such populations. Considering the vast heterogeneity in genetics and ethnicity across Russia’s regions, multigene panels enable the examination of potential genetic contributors to hereditary cancer in a more comprehensive manner. Multigene panels are preferable to traditional single-gene testing, as the latter often misses less common variants or those specific to certain ethnic groups. By contrast, multigene panels cast a wide net and encompass a range of genes linked to different hereditary cancer syndromes.

Multigene panels enable the assessment of multiple genes in one test, simplifying the diagnostic process and providing a comprehensive understanding of an individual’s genetic predisposition to cancer. This efficiency is particularly vital in diverse populations, where varying genetic profiles can result in distinct patterns of hereditary cancer syndromes.

Additionally, multigene panels have versatile utility beyond identifying pathogenic variants in coding regions. They enable the investigation of noncoding intronic regions, unveiling potential regulatory elements that could impact cancer predisposition. This well-rounded approach is especially critical in heterogeneous populations, where distinct genetic variants may contribute to the risk of hereditary cancer.

The application of multigene panels also addresses challenges posed by the clinical overlap of symptoms among various hereditary cancer syndromes. In heterogeneous populations, the diversity of genetic factors contributing to cancer predisposition can result in overlapping clinical presentations. Multigene panels provide a refined diagnostic approach by allowing for the simultaneous analysis of genes linked to distinct syndromes. This improves the accuracy of diagnosis, delivering a thorough comprehension of the genetic factors involved.

### 4.1. Novel Undescribed Variants

In our study, we identified 16 previously undescribed likely pathogenic variants in genes included in our multigene panel (Table 2). Most of these variants (12/16) are nonsense or frameshift variants, which lead to the formation of a premature stop codon. As a consequence, the resulting mRNA of this gene will be degraded by the well-studied mechanism of nonsense-mediated decay (NMD). All of these variants are not present in population genetic databases and previously were not described. Variants were characterized as likely pathogenic according to ACMG criteria. The discovered mutations also include variants located in canonical splicing sites. Variants leading to changes in the canonical splice site nucleotides (±1 or ±2) are referred to as loss-of-function (LOF) variants. Functional studies involving mRNA and protein analysis could confirm the molecular mechanism of pathogenicity.

#### 4.1.1. Mutations in *MLH1* Gene

*MLH1* refers to mismatch repair (MMR) genes, which participate in recognizing and repairing DNA damage. Pathogenic LOF variants in *MLH1* lead to the development of Lynch hereditary cancer syndrome, which is characterized by clinical and genetic heterogeneity [22,23]. Lynch syndrome is recognized for its predisposition to colorectal, endometrial, and various other cancers. This genetic condition is attributed to inherited pathogenic variants present in a heterozygous state within the *MLH1*, *MSH2*, *MSH6*, *PMS2*, and *EPCAM* genes. Cancer risk and survival correlates with mutations in the specific gene responsible for the development of Lynch syndrome. Pål Møller et al. reported cumulative risks at 75 years for various cancers associated with heterozygous mutations in the *MLH1*, *MSH2*, and *MSH6* genes. The findings revealed the following cumulative risks: colorectal cancer—46%, 43%, and 15% for *MLH1*, *MSH2*, and *MSH6* gene mutations carriers, respectively; endometrial cancer—43%, 57%, and 46%; ovarian cancer—10%, 17%, and 13%; upper gastrointestinal cancers—21%, 10%, and 7%; urinary tract cancers—8%, 25%, and 11%; prostate cancer—17%, 32%, and 18%; and brain tumors—1%, 5%, and 1% [24].

Hereditary nonpolyposis colorectal cancer is most common in patients with Lynch syndrome. In our research, pathogenic mutations in *MLH1* were found in four patients, three of whom had colorectal cancer. We discovered the previously undescribed c.160_166del variant in the *MLH1* gene, a 7-bp nucleotide deletion in exon 2, which leads to the formation of a premature stop codon. In our research, two patients with primary multiple tumors were also found to have likely pathogenic undescribed genetic variants in the *MSH2* gene c.893del and c.1729del in a heterozygous state. Both variants also lead to a frameshift and a formation of a premature codon.

#### 4.1.2. Mutations in *EPCAM* Gene

EPCAM (epithelial cell adhesion molecule) is a calcium-independent type I transmembrane glycoprotein. Initially identified as a tumor-associated antigen, EPCAM gained this recognition due to its elevated expression in rapidly proliferating epithelial tumors [25]. Extensive in vitro and in vivo studies have highlighted the critical role of EPCAM in migration, cell adhesion, proliferation, and signaling [26]. Notably, germline mutations in the human *EPCAM* gene have been identified as the underlying cause of congenital diarrhea with tufting enteropathy, a rare autosomal recessive disorder [27]. Ligtenberg et al. found deletions in the 3-prime end of the *EPCAM* gene, resulting in inactivation of the adjacent *MSH2* gene. This inactivation occurred by induction of methylation in the *MSH2* promoter in tissues expressing EPCAM [28]. In addition, Kuiper et al. performed an analysis of 45 Lynch syndrome families with *EPCAM* deletions. These included 27 families identified by targeted genomic screens in cohorts of unexplained Lynch-like families. Currently, it has been shown that 3’ *EPCAM* deletions lead to hypermethylation of the MSH2 promoter, resulting in Lynch syndrome [29]. The underlying mechanism for 3’ *EPCAM* deletion-mediated epigenetic silencing has not yet been clearly established. It is also unclear whether LOF mutations in other regions of the *EPCAM* gene are responsible for the development of Lynch syndrome, including splicing disorder mutations. In our study, we found only one likely pathogenic variant, c.184+1G>A, which is a single nucleotide substitution in the canonical splice donor site of the intron 2 *EPCAM* gene. This variant was found in one patient with breast cancer.

#### 4.1.3. Mutations in *MLH3* Gene

The *MLH3* gene is a member of the MutL-homolog (MLH) family of DNA mismatch repair (MMR) genes. *MLH* genes maintain genomic integrity during DNA replication and after meiotic recombination. Several studies have been conducted on the possible relationship between the presence of germline mutations in *MLH3* and the development of hereditary nonpolyposis colorectal cancer [30,31]. Researchers found no clear relationship between mutations in the *MLH3* gene and the development of colorectal cancer. However, Liu et al. showed in their work that *MLH3* is a low penetrant-risk gene for colorectal cancer. In the observed tumor samples, the presence of *MLH3* mutations did not correspond with microsatellite instability. This suggests a lack of involvement of *MLH3* in carcinogenesis through the disruption of DNA mismatch repair mechanisms [32]. However, Taylor et al. proposed that mutations in *MLH3* might be implicated in the pathogenesis of certain endometrial cancer cases [33]. In our study, only one likely pathogenic frameshift variant c.1544del in the *MLH3* gene was detected in a patient with ovarian cancer. Honglin Song et al. performed a large study to investigate associations of common variants in *MMR* genes, including *MLH3* and ovarian cancer, using a single nucleotide polymorphism tagging approach [34]. They concluded that two common variants, rs7303 and rs175080, are unlikely to cause ovarian cancer [34]. The relationship between germline mutations in *MLH3* and ovarian cancer risks remains unclear.

#### 4.1.4. Mutations in *ATM* Gene

The ATM serine/threonine kinase (ATM) is a member of the phosphoinositide 3-kinase-related protein kinase (PIKK) family and plays a critical role in the DNA damage response [35]. Pathogenic loss-of-function (LOF) variants in the *ATM* gene are responsible for ataxia–telangiectasia, a rare autosomal recessive disorder characterized by neurodegeneration, increased sensitivity to radiation, immunodeficiency, and a predisposition to cancer [36,37].

Individuals who are heterozygous carriers of pathogenic germline variants in *ATM* are at increased risk of developing several types of cancer. This increased susceptibility includes hematopoietic, breast, pancreatic, and gastric cancers [38,39,40]. The 2-bp nucleotide deletion c.2227_2228del in exon 14 and 5-bp nucleotide insertion c.6060_6064dup in exon 41 that we detected both result in a frameshift, which causes the formation of a premature stop codon. The variants were found in patients with pancreatic cancer and breast cancer (Table 2). Fang-Chi Hsu et al. showed results that the cumulative risk of pancreatic cancer among individuals with a germline pathogenic *ATM* variant was estimated to be 1.1% (95% CI, 0.8–1.3%) by age 50 years; 6.3% (95% CI, 3.9–8.7%) by age 70 years; and 9.5% (95% CI, 5.0–14.0%) by age 80 years [41]. Neha Nanda et al. also described the role of *ATM* germline mutations in the development of pancreatic cancer [42]. Their research demonstrated a correlation between *ATM* variants and the susceptibility to breast cancer [43]. Based on seven adjusted case-control studies, the odds ratio (OR) for this association was calculated to be 1.67 (95% CI: 0.73–3.82). In nine unadjusted case-control studies, the crude OR was 2.27 (95% CI: 1.17–4.40), and in two cohort studies, the relative risk (RR) was estimated at 1.68 (95% CI: 1.17–2.40) [43].

#### 4.1.5. Mutations in *GALNT12* Gene

The *GALNT12* gene encodes a member of the UDP-GalNAc:polypeptide N-acetylgalactosaminyltransferase family. These enzymes play a critical role in catalyzing the transfer of N-acetylgalactosamine (GalNAc) from UDP-GalNAc to a serine or threonine residue on a polypeptide acceptor. This process marks the first step in O-linked protein glycosylation [44]. In their study, Guda et al. suggested that germinal LOF variants in *GALNT12* lead to increased susceptibility to colorectal cancer [45]. Further clinical studies have shown a correlation between pathogenic mutations in *GALNT12* and colorectal cancer [46]. In our cohort, we identified a 14-bp nucleotide deletion c.171_184del in *GALNT12* in a patient with breast cancer. The potential impact of LOF mutations in the *GALNT12* gene on breast cancer risk has not yet been studied. However, Banu Arun et al., in their multi-gene panel testing of breast cancer patients, also found pathogenic germline variants in *GALNT12* [47]. We hope that our study and others will contribute to a more thorough investigation about the relationship between mutations in the *GALNT12* gene and cancer susceptibility.

#### 4.1.6. Mutations in *MUTYH* Gene

The *MUTYH* gene encodes a base excision repair DNA glycosylase that helps protect cells against the mutagenic effects of guanine oxidation [48]. A series of clinical observations have shown that biallelic and heterozygous germline pathogenic variants in *MUTYH* are probably associated with the development of familial adenomatous polyposis [40,41,42,43,44,45,46,47,48,49,50,51]. Farrington and colleagues conducted a comprehensive study revealing that biallelic *MUTYH* mutations result in a 93-fold increase in the risk of colorectal cancer [52]. For heterozygous carriers, there was also a 1.68-fold increased risk for those over the age of 55 years. In our work, P/LP heterozygous mutations were found in three patients with breast cancer, two patients with pancreatic cancer, one with ovarian cancer, and one with colorectal cancer (Table 1). Some studies demonstrate an increased risk of breast cancer in patients with pathogenic mutations in *MUTYH* [53,54,55], but large international clinical trials have not yet been conducted. We found a previously undescribed likely pathogenic c.705G>T (p.Trp235Cys) *MUTYH* variant in a patient with pancreatic cancer. The detected variant is a missense substitution in the first nucleotide of exon 10. It results in the amino acid replacement of tryptophan at position 235 to cysteine. Replacing an aromatic amino acid to an aliphatic sulfur-containing one in a protein may lead to disruption of its function. The close position of this missense variant to the splice acceptor site of intron 9 may lead to splicing disruption, as confirmed by in silico splicing prediction tools [56]. Germline *MUTYH* mutations have previously been identified in patients with pancreatic cancer [57,58]; however, further clinical studies are necessary to determine the risks in cancer development.

#### 4.1.7. Mutations in *POLE* Gene

The *POLE* gene encodes the central catalytic subunit of DNA polymerase epsilon, one of the four nuclear DNA polymerases in eukaryotic cells, which is involved in DNA repair [59]. Biallelic pathogenic genetic variants in *POLE* lead to the development of autosomal recessive diseases: FILS syndrome (OMIM #615139) and IMAGE-I syndrome (OMIM #618336) [60,61,62]. In the study by Claire Palles et al., it was first identified that heterozygous variants in the *POLE*, which lead to disruption of the exonuclease domain, cause an increased risk of colorectal cancer development [63]. Further clinical studies confirmed this relationship and also identified many different pathogenic variants in *POLE* [64,65]. We reported three novel likely pathogenic mutations in *POLE*: c.802-2A>G, c.6665_6666del, and c.799C>T in patients with colorectal cancer, ovarian cancer, and pancreatic cancer, respectively. Likely pathogenic variants c.6665_6666del and c.802-2A>G are a frameshift deletion and a single nucleotide substitution in the canonical splice site, respectively. The variant c.799C>T (p.Pro267Ser) is a missense that is predicted to disrupt splicing according to in silico prediction tools [56]. All variants we found potentially lead to a loss of function in the exonuclease domain or the whole protein. However, it is worth performing functional mRNA studies to confirm the molecular mechanism of pathogenicity in c.802-2A>G, c.6665_6666del, and c.799C>T. Cases of extracolonic tumors have been reported, including endometrial, ovarian, pancreatic tumors [66,67]. In study by Pilar Mur et al., it was shown that pathogenic germline mutations in the *POLE* and *POLD1* genes most commonly associated with colorectal, endometrial and ovarian cancer tumor types [68]. 

NGS has not only facilitated the identification of known cancer-associated mutations, but has also played a critical role in the discovery of novel undescribed variants. By sequencing large numbers of genes, NGS enables researchers to identify previously unknown genetic alterations that may contribute to the development of cancer. These novel variants can provide valuable insights into the underlying mechanisms of tumorigenesis and potentially uncover new therapeutic targets. The ability of NGS to detect and characterize these novel variants has greatly expanded our understanding of cancer genetics and holds great promise for advances in cancer diagnosis and treatment.

## 5. Conclusions

Our study contributes valuable genetic data to the field of hereditary cancer syndromes, particularly in the context of the Russian population. The identification of novel pathogenic variants and their associations with specific cancer types facilitates further research and underscores the importance of comprehensive genetic testing in clinical practice. These insights have significant implications for the development of personalized approaches to cancer prevention and treatment. As we analyze the complex genetic data we have collected, our research aims to make a significant contribution to the worldwide comprehension of hereditary cancer syndromes. At the same time, we strive to address the specific needs and challenges presented by the genetic diversity found within the Russian population. Our goal is to lay the groundwork for more effective and individualized methodologies for the prevention and management of hereditary cancers.

## Figures and Tables

**Figure 1 cancers-16-00085-f001:**
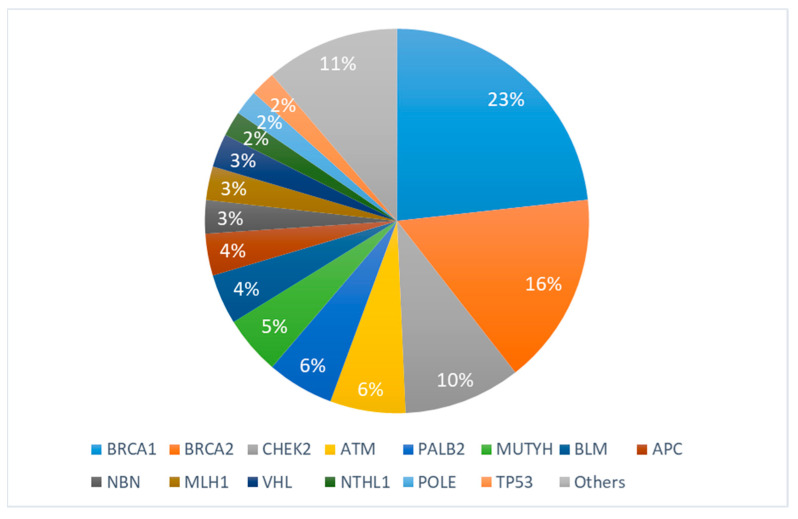
Spectrum of identified genetic variants. “Others” include the following genes: *PMS2*, *MSH*, *BARD1*, *MSH2*, *MEN1*, *MSH6*, *CDKN2A*, *EPCAM*, *TSC2*, *GALNT12*, *MLH3*, and *BRIP1*.

**Table 1 cancers-16-00085-t001:** Frequency of P/LP variants among tested individuals.

Genes	Gastric Cancer	Colorectal Cancer	Pancreatic Cancer	Breast Cancer	Ovarian Cancer	Multiple Primary Tumors	P/LP	Sum	Gender (Male/Female)	Age of Manifestation (Mean ± SD)
*BRCA1*			1	24	7	1	30/3	33	2/31	43.9 ± 11.2
*BRCA2*			2	14	5	2	21/2	23	1/22	48.4 ± 11.6
*CHEK2*		1	1	11	1		8/6	14	2/12	42.5 ± 12.4
*ATM*	1		3	5			7/2	9	2/7	43.6 ± 10.2
*PALB2*	1			7			6/2	8	2/6	48.6 ± 8.2
*MUTYH*		1	2	3	1		3/4	7	2/5	41.6 ± 10
*BLM*				5		1	7/0	6	2/5	44 ± 12.7
*APC*		3		2			5/0	5	4/5	37.1 ± 14.7
*NBN*	1			2		1	4/0	4	1/3	57 ± 8
*MLH1*		3				1	3/1	4	0/4	39.2 ± 6.1
*VHL*			1	3			0/4	4	0/4	41 ± 12.6
*NTHL1*		1		1	1		3/0	3	0/3	46.6 ± 6
*POLE*		1	1	1			0/3	3	0/3	32.3 ± 12.5
*TP53*				3			2/1	3	0/3	37.6 ± 15.9
*PMS2*		1	1				2/0	2	2/0	49 ± 5.6
*MSH3*		1	1				2/0	2	1/1	37.5 ± 3.5
*BARD1*				2			2/0	2	0/2	44 ± 1.4
*MSH2*					1	1	1/1	2	1/1	50 ± 4.2
*MEN1*	1						0/1	1	1/0	47
*MSH6*		1					0/1	1	1/0	31
*CDKN2A*			1				1/0	1	1/0	62
*EPCAM*				1			0/1	1	0/1	49
*TSC2*				1			0/1	1	1/0	42
*GALNT12*				1			0/1	1	0/1	40
*MLH3*					1		0/1	1	0/1	54
*BRIP1*					1		1/0	1	0/1	37

**Table 2 cancers-16-00085-t002:** Novel undescribed variants.

Gene	Transcript	Chromosomal Change	Coding	Protein	ACMG	Diagnosis
*ATM*	NM_000051.4	chr11:108256317delTC	c.2227_2228del	p.Ser743ArgfsTer21	LP	Pancreatic cancer
		chr11:108315875insGCTGT	c.6060_6064dup	p.Gly2022AlafsTer27	LP	Breast cancer
*BRCA1*	NM_007294	chr17:43070934insT	c.4980dup	p.Glu1661ArgfsTer18	LP	Breast cancer
		chr17:43094515delTT	c.1015_1016del	p.Lys339GlyfsTer6	LP	Breast cancer
*BRCA2*	NM_000059	chr13:32336925insTT	c.2570_2571insTT	p.Arg858Ter	LP	Pancreatic cancer
		chr13:32340800delATTA	c.6446_6449del	p.Ile2149LysfsTer18	LP	Breast cancer
*EPCAM*	NM_002354	chr2:47373571G>A	c.184+1G>A	-	LP	Breast cancer
*GALNT12*	NM_024642	chr9:98807865delCGCGCCCCGGGCGG	c.171_184del	p.Pro58AlafsTer42	LP	Breast cancer
*MLH1*	NM_000249	chr3:36996662delGGAGGCC	c.160_166del	p.Gly54Ter	LP	Colorectal cancer
*MLH3*	NM_001040108	chr14:75048112delG	c.1544del	p.Pro515HisfsTer11	LP	Ovarian cancer
*MSH2*	NM_000251	chr2:47414369delA	c.893del	p.Gln298ArgfsTer3	LP	Ovarian cancer
		chr2:47471032delA	c.1729del	p.Ile577LeufsTer13	LP	Multiple primary tumors
*MUTYH*	NM_001048174	chr1:45332310C>A	c.705G>T	p.Trp235Cys	LP	Pancreatic cancer
*POLE*	NM_006231	chr12:132624986delCA	c.6665_6666del	p.Leu2222GlnfsTer81	LP	Ovarian cancer
		chr12:132676655T>C	c.802-2A>G	-	LP	Colorectal cancer
		chr12:132677365G>A	c.799C>T	p.Pro267Ser	LP	Pancreatic cancer

LP—likely pathogenic.

## Data Availability

The data are not publicly available due to restrictions; these data contain information that could compromise the privacy of research participants. Requests to access the additional data should be addressed to the following email: s.nikolaev@mknc.ru.

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
