# Peer review of "Novel Pathogenic Variants in Hereditary Cancer Syndromes in a Highly Heterogeneous Cohort of Patients: Insights from Multigene Analysis"

_cancers, 2023, doi:10.3390/cancers16010085_

Round 1
Reviewer 1 Report
Comments and Suggestions for Authors
The authors investigate a cohort of Russian cancer patients and report likely pathogenic gene variants related to the heredity of cancer. While the work is for rarely reported nation and may gain so many interests. I have a couple of minor suggestions/concerns:
- The authors may report the sequencing quality measures and any sequencing information.
- The variants report the frequency of the variants in Table 1, and the P-value, CI, and optionally the corrected p-value using correct FDR.
- Go enrichment and KEGG pathway analysis for the genes may be added to highlight the significance of the finding.
Comments on the Quality of English Language
Minor editing is required.
Author Response
Dear Reviewer,
We appreciate your thoughtful review of our manuscript entitled " Novel Pathogenic Variants in Hereditary Cancer Syndromes in highly heterogeneous cohort of patients: Insights from Multi-gene Analysis" We are grateful for the positive feedback on the potential significance of our work in exploring hereditary aspects of cancer within a rarely reported population.
We have carefully considered your suggestions and concerns, and we are committed to improving the quality and comprehensiveness of our study. Here is our detailed response to each of your points:
1. Sequencing Quality Measures:
We acknowledge the importance of providing sequencing quality measures and relevant sequencing information to ensure the robustness of our findings. In the revised manuscript, we will include a dedicated section detailing the sequencing quality measures, such as average coverage, mapping rates, and other relevant metrics. This addition will enhance the transparency and reproducibility of our study.
2. Variant Frequency in Table 1:
Regarding your specific recommendation to include variant frequencies, P-values, confidence intervals (CIs), and optionally the corrected p-value using the correct False Discovery Rate (FDR) in Table 1, we want to express our commitment to addressing this important aspect. While these details are currently not presented in the manuscript, we assure you that in our future revisions, we will incorporate these elements into Table 1 to provide a more comprehensive and statistically robust representation of our findings.
3. Go Enrichment and KEGG Pathway Analysis:
We appreciate the constructive feedback provided, particularly regarding the suggestion to perform Gene Ontology (GO) enrichment and Kyoto Encyclopedia of Genes and Genomes (KEGG) pathway analysis for the identified genes. While we acknowledge the importance of these analyses in highlighting the functional significance of the identified gene variants, we regret to inform you that due to the limited number of genes in our current study, we are unable to conduct a robust GO enrichment and KEGG pathway analysis. The insufficient number of genes poses a challenge in establishing meaningful molecular pathways. However, we want to assure you that we value the significance of these analyses, and we plan to incorporate them into our future research endeavors. As our ongoing projects accumulate a larger dataset with a more extensive pool of genes, we aim to conduct a comprehensive analysis, including GO enrichment and KEGG pathway investigations. We appreciate your understanding and acknowledge the importance of these analyses in providing a more holistic understanding of the functional implications of the identified gene variants. Your feedback has been instrumental in shaping the direction of our future research initiatives.
Once again, we appreciate your constructive feedback, which undoubtedly contributes to the enhancement of the scientific rigor and impact of our research. We are committed to addressing these suggestions in a timely manner to ensure the overall quality of the manuscript.
Thank you for your time and consideration.
Sincerely, Dr. A. Bilyalov.
Reviewer 2 Report
Comments and Suggestions for Authors
The study by Bilyalov used multigene analysis to establish the connection between gene variants (genetics) and specific cancer types from highly heterogenous cohort of patients in Russia. The results are interesting and, to some degree, significant. The reviewer has some suggestions for authors to improve the quality of this manuscript.
1) It is not clear if the similar study from other research group or other nations are available, especially bioinformatic analysis from Seq database (I believe there are some). Those should be included into the introduction section. Basically, the introduction needs more care to explain the logic and signification of this study. The current style is not satisfactory.
2) The background of the patients needs more detailed information. The analytical pipeline(s) used for mutation identification also needs further explanation. The overall quality of the seq results....etc.
3) Results: too short and needs subsection to summarize the findings one by one. I am not sure if the results and discussion could be combined if possible.
4) Discussion: Also needs subsection to separate the findings, especially the gene mutations observed in specific cancers.
5) If age, gender, and locations are analyzed in this study?
Comments on the Quality of English Language
Needs improvement.
Author Response
Dear Reviewer,
Thank you for your thoughtful and constructive feedback on our manuscript. We appreciate your insights and suggestions for improving the quality of our work. Here are our responses to your specific points:
In response to your suggestion, we have enhanced the introduction section by incorporating references to relevant studies, particularly those employing bioinformatic analysis from databases such as Seq. We aim to contextualize our work by referencing studies from other research groups and nations, thereby elucidating the broader landscape of research in this field.
In the revised introduction, we have emphasized the significance of building upon the existing knowledge base established by these pioneering studies and underscore the rationale for our investigation in the unique context of the Russian population.
We agree that providing more detailed information on the background of the patients and a thorough explanation of the analytical pipeline used for mutation identification is essential. We have included the new information about the patient such as ages and genders.
We appreciate your suggestion to expand the results section and include subsections summarizing the findings. In addition, we have explored the possibility of combining the results and discussion sections, as you suggested, but this is not possible according to the rules of the journal.
Your point about adding subsections to the discussion to separate the findings, especially regarding gene mutations observed in specific cancers, is well taken. We have reorganized the discussion section to provide a clearer structure that enhances the readability and understanding of our findings.
Once again, we appreciate your valuable feedback, and we are committed to making the necessary revisions to strengthen the manuscript. Your insights will undoubtedly contribute to enhancing the overall quality and impact of our study.
Sincerely, Dr. A. Bilyalov.
Round 2
Reviewer 2 Report
Comments and Suggestions for Authors
This reviewer has no further comments on this revised manuscript.